# Isothermal Detection Methods for Fungal Pathogens in Closed Environment Agriculture

**DOI:** 10.3390/jof10120851

**Published:** 2024-12-10

**Authors:** Aylwen Cotter, Peter Dracatos, Travis Beddoe, Kim Johnson

**Affiliations:** 1Australian Research Council Industrial Transformation Research Hub for Medicinal Agriculture, Bundoora 3083, Australia; w.cotter@latrobe.edu.au (A.C.); t.beddoe@latrobe.edu.au (T.B.); 2La Trobe Institute for Sustainable Agriculture and Food, Department of Ecological, Plant and Animal Sciences, La Trobe University, Bundoora 3083, Australia; p.dracatos@latrobe.edu.au

**Keywords:** pathogenic fungus, closed environment agriculture, molecular methods, detection, isothermal, LAMP

## Abstract

Closed environment agriculture (CEA) is rapidly gaining traction as a sustainable option to meet global food demands while mitigating the impacts of climate change. Fungal pathogens represent a significant threat to crop productivity in CEA, where the controlled conditions can inadvertently foster their growth. Historically, the detection of pathogens has largely relied on the manual observation of signs and symptoms of disease in the crops. These approaches are challenging at large scale, time consuming, and often too late to limit crop loss. The emergence of fungicide resistance further complicates management strategies, necessitating the development of more effective diagnostic tools. Recent advancements in technology, particularly in molecular and isothermal diagnostics, offer promising tools for the early detection and management of fungal pathogens. Innovative detection methods have the potential to provide real-time results and enhance pathogen management in CEA systems. This review explores isothermal amplification and other new technologies in detection of fungal pathogens that occur in CEA.

## 1. Introduction

Fungi are a large group of eukaryotic organisms, with a diverse range of taxa that play essential roles in the environment and many human activities [1]. These include, but are not limited to, aiding with the decomposition of organic matter and the production of antibiotics and sources of food [2]. Other fungal species can contaminate or cause toxicity in ingested plant-based foods or medicines [3]. Fungi can also be pathogenic and cause disease in both animals and plants. Some of the most destructive diseases of crops are caused by fungal pathogens. It has been estimated that around 15,000 fungal species cause between 70–80% of all plant diseases [4]. For example, wheat is the most widely produced food crop globally but is the primary host for three different rust pathogen species from the genus *Puccinia: P. striiformis* f. sp. *tritici* (stripe rust), *P. graminis* f. sp. *tritici* (stem rust) and *P. triticina* (leaf rust). Yield losses can vary, up to 100% for stripe rust, 90% for stem rust and 70% for leaf rust, leading to major yield losses, food insecurity and the disruption of global markets [5,6,7,8]. Pest and diseases can cost food producers billions of dollars each year, with countries like the Unites States of America estimating losses of USD 21 billion through prevention techniques such as insecticides, fungicides, and/or crop losses [9].

Increased urbanization, land degradation and limited resources are reducing the amount and quality of arable land for agriculture, along with the additional challenges of climate change [10]. Indoor crop growing systems, termed closed or controlled environment agriculture (CEA) and protected cropping environments (PCEs), are proposed as one of the solutions to maintain sustainable food production [10]. Indoor agriculture designs have the potential to reduce the overall environmental footprint with approaches such as recycled water, renewable energy, and urban locations to reduce transport [10].

CEA control growth conditions such as temperature, watering and nutrient supply, leading to increased productivity and yields of greater than 50% when compared to field yield [11]. In CEA systems such as greenhouses and vertical farms, approaches to mitigate the introduction of pests and disease include manual or automated sanitisation systems. To reduce the presence of fungi in these environments, methods such as increasing air flow using fans, the use of humidifiers to encourage plant health, removing infected plants and the use of beneficial insects such as fungal gnats that feed on fungi in soils aim to keep the plants healthier and reduce the likelihood of infection. PCEs are similar to closed environments, however, they allow air to flow freely from one end of the structure to the other, which can lead to the infestation of pests and diseases [12]. It is challenging to exclude all pathogens entirely from PCEs as they are often introduced through watering systems, soils, fertilisers, footwear, clothing and seeds. For example, the oomycete pathogen *Phytophthora cryptogea* that causes a root disease in hydroponic lettuces was shown to infect seedlings grown repeatedly on the same site through the watering systems [13]. Disease outbreaks are shown to occur more frequently in CEAs when plants become stressed, reducing the effectiveness of the plant immune response leading to increased fungal colonisation [14]. There is currently limited information about the amount of crop loss and associated costs to indoor cropping industries as a result of fungal disease-causing pathogens. CEA losses from the Australian hydroponic lettuce industry were recorded to be between 20 and 30% for root diseases alone, with complete crop losses occurring during the warmer months [13]. Diseases such as the tomato brown rugose virus can result in 100% crop loss in CEA, emphasizing how damaging exotic pathogens can be to growers [15].

There are several crop types that are currently being grown in CEA including: vegetables, cereals, flowers, herbs, fruits and medicinal crops [16]. Cereal crops such as rice grown indoors have been shown to be infected by *Magnaporthe grisea* that causes rice blast disease, which can infect young seedlings, killing the plant by infecting nodes, and stems causing grain loss [17,18]. Fruits such as strawberries, melons and blueberries are prone to *Botrytis cinerea* infection in CEA, and herbs such as oregano, basil and mint are prone to *Fusarium* spp. infection [16]. When grown indoors, medicinal crops such as cannabis and aloe vera have been reported to show diseases such as *Alternaria alternata* and *Fusarium* spp. [19,20]. The indoor medicinal cannabis industry is frequently affected by the necrotrophic fungus *Botrytis cinerea* through its ability to infect susceptible cultivars both at post-harvest or whilst in storage [21]. The ornamental flower industry that includes roses and carnations are prone to powdery mildews and *Fusarium* spp. [22].

In order to prevent crop damage and cost to the indoor agriculture industry, rapid and early detection methods of pathogens can assist with strategies for control and prevention of further disease spread. Traditional methods of fungal identification in agriculture have been through the visualization of signs or symptoms on plants based on morphological and/or taxonomic features. Molecular approaches are increasingly being adopted to detect fungal presence prior to disease development [23]. In a controlled environment setting, the use of environmental (eDNA) sampling can be adopted to aid in the obtaining of DNA samples for molecular processing. eDNA sampling targets spores moving through the air or run off from watering systems. Unlike broadacre agriculture settings where fungal contaminants from birds, insects, faecal matter and other flora can complicate eDNA sampling, indoor areas are suitable for targeted disease detection. Furthermore, combining more than one method of detection can lead to point of care diagnostics. Point of care diagnostics is the carrying out of tests using specific devices or kits next to or near the patient in health settings, or in field for the agriculture industry in a quick and efficient manner without the need to send the samples off to a laboratory [24,25]. In the following sections we outline fungal pathogens common to the indoor agriculture setting and molecular approaches used for detection.

## 2. Pathogenic Fungi Affecting CEA

*Botrytis* and powdery mildew causing pathogens (*Golovinomyces* spp. and *Blumeria* spp.), *Pythium*, *Sclerotinia*, downy mildew (*Peronospora* spp.) and *Fusarium* spp. are the most prevalent fungi affecting CEA industries (Figure 1) [21,26]. Some are host specific, such as the powdery mildew, and others such as *Botrytis cinerea* have a broad host range [19,27]. Diseases caused by fungal pathogens can infect plants at every stage of growth and development however infection depends on many factors including conditions, age, stress, genetics and the fungal species itself [28]. Some species of *Botrytis* and *Fusarium* can cause health issues in humans if consumed through exposure to spores or mycotoxins [19,29].

### 2.1. Botrytis *sp.*

There are approximately 35 species of Botrytis [31]. *Botrytis cinerea* is one of the most prominent and destructive pathogens with a broad host range comprising approximately 200 different plant species [32]. Hosts include fruits such as strawberries and tomatoes, and cut flowers such as roses [33,34]. *Botrytis* was first described in 1794 by Christiaan Hendrik Persoon, and can sometimes be confused with *Sclerotinia* spp. [32,35]. *B. cinerea* is distributed all over the world and infection can be present without the appearance of symptoms [36]. This fungal species can lay dormant for days to months and when conditions are of high humidity with cool to moderate temperatures, a grey mould starts to form on infected plants (Figure 1B) [37]. *Botrytis* releases spores, known as conidia, and a distinguishing feature of *Botrytis* is the oval shaped conidia with colonies of dark grey and aerial mycelium [38]. During infection, *Botrytis* conidia released from the conidiophore into the air and land on the hosts surface. The initiation and formation of appressorium subsequently leads to penetration and infection of plant tissue at points of damage or natural openings such as the stomata [32]. Depending on the plant host, the type of plant tissue that is infected varies. For example, on fruiting plants the site of infection is generally the floral organs [32]. Studies have determined that fungal pathogens such as *B. cinerea* can grow asymptomatically within the host subverting the plant defence response prior to switching to killing the host cells to feed on dead cells through acquisition of nutrients as part of its necrotrophic lifestyle [39]. Multiple virulence mechanisms, such as the use of effector proteins, toxins and lytic enzymes and the secretion of oxalic acid to reduce pH in host tissue are involved in infection [34]. *Botrytis* spores can also induce an immune response in humans that can affect health and are known as allergens [19]. Although air is a major form of transmission, the conidia can also travel through watering systems. *B. cinerea* is considered a disease of agricultural importance due to its broad host range leading to widespread crop losses especially to the horticultural industry and tendency to develop fungicide resistance [32].

### 2.2. Powdery Mildew

Over 900 species of powdery mildew and even more sub-species have been described [40,41]. In contrast to *Botrytis* spp., mildews are an obligate biotrophic fungus meaning they obtain their nutrients from intact living cells and require living plant host tissue to complete their lifecycle [41]. Closely related powdery mildew species, such as *Golovinomyces* and *Blumeria* spp. are difficult to distinguish morphologically, with the main differences being host range and conidial and hyphal structures, which can only be seen under a microscope [41,42]. As the name suggests the symptoms mimic small white powder patches, commonly found on leaves but can be found on stems and buds and can cover large amounts of plant tissue (Figure 1J) [43]. Powdery mildew affects a diverse array of crops such as grapes, wheat, barley, oat, cannabis and some fruit and vegetable species. The pathogen acts as a sink for carbon and micronutrients causing the depletion of nutrients and carbon that in turn causes yield and grain quality to decrease [44]. Powdery mildew species reproduce mainly via asexual production of conidiospores. The spore lands on the plant host and the appressorium produces sufficient pressure to lyse the cuticle of the epidermal cell layer [45]. Successful colonisation of the epidermal cell by the powdery mildew fungus is characterised by the production of haustoria which facilitates the acquisition of nutrients from the host and in parallel the secretion of effectors by the pathogen [46].

### 2.3. Pythium

*Pythium* spp. were first discovered in 1858 by Pringsteim and can affect animals and humans depending on species [47,48]. *Pythium* spp. cause root rot or dampening off resulting in yield losses to many varieties of vegetables and ornamental crops including corn, potato, wheat, oats, tobacco and peanuts to name a few [49,50]. On plants, the infection develops into necrotic lesions on the roots (Figure 1H), that ends up affecting the stem and leaves, by producing hyphae that extract nutrients from the host [51,52]. *Pythium* spp. use carbohydrate active enzymes (CAZymes) that aid in the penetration of the plants cell walls and assists in further disease establishment [52]. Other symptoms can include reduced growth, black or brown discolouration visible on seeds and young roots as well as wilting of young or juvenile tissue such as in seedlings [52]. *Pythium* can contaminate irrigation water systems as well as soilless systems and can also be spread through gnats [53]. There are many species of *Pythium* that have been identified that are saprophytic and are not plant pathogens [50]. *Pythium* prefers environmental conditions that are moist with high temperatures, with some species having the ability to remain dormant for long periods of time [50]. Effects on the plants depend on the species infecting them. Some species such as *P. spinosum*, *P. arrhenomanes*, *P. myriotylum*, and *P. dissotocum* affect the seeds, while those causing issues such as dampening off for seedlings and root rot in more mature plants include *P. afertile*, *P. arrhenomanes*, *P. dissotocum*, *P. elongatum*, and *P. spinosum* [54,55,56].

### 2.4. Sclerotinia

*Sclerotinia* is a necrotrophic fungus with a wide host range, infecting >450 plant species including but not limited to: soybean, rapeseed, onion, garlic, canola, sunflower and, more recently, cannabis [57,58,59]. The diseases caused by either *S. sclerotium* or *S. minor* infection are deemed of high economic importance due to the vastness of the hosts infected, and the damage caused by the diseases [57]. Infection causes stem rot and cankers, which can then turn to shoot necrosis. *Sclerotinia* is often referred to as white mould, though has many other names such as crown rot and cottony rot [60]. The infection begins with pale or dark brown lesions and develops into white mycelium that look like cotton wool (Figure 1D). The hyphae then develop small black balls called sclerotia [57]. The sclerotia are renowned for staying dormant in the soil ecosystems for >10 years providing additional challenges for controlling the disease during favourable environmental conditions [61]. The disease cycle is characterised by ascospores landing on host plant tissues and germination begins causing infection preferentially in temperate conditions [62].

### 2.5. Other Notable Fungal Pathogens

Other pathogens that can occur in CEA include downy mildew, caused by pathogens such as *Plasmopara viticola* and *Hyaloperonospora arabidopsidis*, as well as *Fusarium* species. Downy mildew affects a variety of agriculture crops including brassicas, cucurbits, lettuce, peas, grapevine and tobacco [63]. The disease creates mosaic-like patterns on the leaves of a susceptible host and can cause 40–90% yield loss in field given favourable temperature and humidity. Infection is caused by oomycetes that germinate via oospores [64]. The genus and species of this disease depends on the host plant it infects.

Diseases caused by *Fusarium* spp. are estimated to be around 1500 species and therefore signs and symptoms can vary depending on the infected host [65]. Not all species of *Fusarium* are pathogenic, however, some can produce mycotoxins that can contaminate food and harm humans and animals [29]. *Fusarium* spp. can effect a variety of agricultural crops including cabbage, tomato, cotton, spinach, peas and cereal grain crops [66]. For example *Fusarium oxysporum* (Figure 1F) have further evolved *formae speciales ff.* spp. based on the ability to infect a specific range of closely related host plant species and cause wilt disease [66]. Specific examples include *F. oxysporum f.* sp. *lypersici* (tomato) and *F.o f.* sp. *vasinfectum* that causes wilt in cotton [67]. For some *F. oxysporum f.* spp. such as *F. oxysporum f.* sp. *conglutinans* that effect brassica and Arabidopsis plants, are influenced by genetic diversity that leads to phenotypic variations. These variations can be shaped by environmental conditions significantly impacting the interactions with host plants; this, in turn, causes natural accessions within a species and genus and only certain *formae speciales* will affect these accessions [68,69].

## 3. Fungal Identification and Treatment in Agricultural Settings

Fungal disease identification in agriculture settings has largely been through visualization of disease signs or symptoms. Regional networks can then be involved in alerting other growers to the occurrence of disease in the area. In addition, weather warning systems make growers aware of climatic conditions that favour disease occurrence, such as increased rainfall and humidity. To date, there is limited information and capability to quantify with any accuracy the prevalence and occurrence of specific fungal pathogens over a growing period. Closed environments may limit some of the complexity by maintaining a relatively stable environment so that the use of diagnostics can be used to identify when pathogens are most likely to occur during plant growth cycle and take preventative action.

Fungal identification and detection methods encompass a range of techniques, including traditional microscopy, culture-based approaches, air and water sampling, chemical treatments and molecular tools such as PCR and isothermal assays, in order to enable accurate identification and monitoring of fungal species in various environments. Historically the identification and diagnosis of fungal pathogens was determined by Kochs postulates, a process of identifying the causal relationship between microbe and disease [70,71]. More recently, identification has been achieved through the observation of different pathogen morphology, culturing to determine lifecycle characteristics, microscopic examination and, then, through the development of DNA sequencing and molecular identification [72,73,74]. The first fungus-like organism to be investigated using a light microscope was the oomycete pathogen *Phytophthora infestans*, the causal agent of potato blight. Detected on potatoes in the 1840s, the strain of *P. infestans* that caused the great potato famine has been traced using DNA analysis [72,75,76].

The disease triangle outlines the relationship between the environment, the plant host and the pathogen [77]. This approach has been used as a method of predicting epidemiological outcomes since the 1960s [77]. More recently there is a fourth aspect to consider, the human effect or society, as well as a fifth element, time. The human effect is a concept based on human potential for transporting pathogens across borders, excessive use of fungicides and plant monocultures that lead to the breakdown in the genetic resistance [77]. Time refers to the event of disease development and the length of time it takes for the disease to grow or latency period (LP). LP, or the time taken for the infection cycle to develop, is affected by numerous factors including seasonal environmental conditions, stage of growth and genetic resistance status in the host plant [77].

Once disease is detected, growers must decide what action to take, and this can include fungicide application, integrated pest management and/or plant destruction [78]. Information through government websites and reputable agriculture associations provide farmers with the latest information and guides to aid in pest and disease management. To validate the identity of the pathogen growers, agronomists and plant pathologists can send samples to a laboratory for accurate identification if needed. In some cases, it is important to notify the appropriate authorities as some pests and pathogens breach biosecurity protocols and are considered exotic incursions, that could potentially have large consequences, this is different depending on the location. The use of point of care (POC) diagnostics for pest and pathogen testing is limited in field settings due to initial development costs and commercialisation, determining which device is best for which pest or pathogen, the ability to sufficiently validate the results and the fact tests are developed to target a few specific pathogens [79,80,81,82]. However, once developed, POC diagnostics can rapidly and cheaply be performed in field with the ability to obtain results in a shorter timeframe than current pathology laboratories [83].

More recent non-destructive approaches include the use of biosensors and imaging for diagnostic detection of fungal disease prior to the appearance of symptoms [84]. Such methods can complement integrated pest management plans. These approaches have been covered in reviews such as [85,86,87]. This review focusses on the molecular approaches such as, but not limited to, Loop-Mediated Isothermal Amplification (LAMP), Helicase-Dependent Amplification (HDA), Whole Genome Amplication (WGA), Nucleic Acid Sequence-Based Amplification (NASBA), Strand Displacement Amplification (SDA), Recombinase Polymerase Amplification (RPA) and Rolling Circle Amplification (RCA) (Table 1). Fungal sporulation is the most efficient detection stage/target due to the period of time it takes for the spores to be released which is defined as the latency period of infection. The development of early detection methods for the sporulation stage of pathogenic fungal growth can reduce the necessity for fungicides and other potentially harmful methods of fungal eradication, as well as potentially reducing the likelihood of human allergic responses and contamination of food and medicinal products. Air detection methods utilize the sampling of spores from air using equipment such as the SKC Biosampler, a liquid cyclonic impinging device that captures spores in a buffer which can then be used to extract nucleic acids for further diagnostic applications [88]. These methods are being adapted for the detection of fungal spores and for DNA testing to determine pathogen genera and species [89]. For outdoor sampling, changes to air sampler design and type, such as the impaction spore trap (BioScout, Marrickville, New South Wales, Australia), have been effective for broadacre agricultural crops [90]. eDNA is increasingly being used to determine microbial biodiversity in a range of settings, such as soils and air [91]. The principles of eDNA sampling could be transferred to the agriculture industry [92]. There is limited published information on fungal in-field sampling in agricultural settings. Most of the current research is aimed at the medical industry focussing on fungal pathogens causing human infection [93,94].

Water detection methods for closed environment agriculture have been used for the detection of root diseases such as those caused by *Fusarium* and *Pythium* spp. in hydroponic lettuces [13]. Current methods test the growing mediums used for the propagation of cuttings, such as coco fibre [21]. Coconut fibre and Rockwool are commonly used growing mediums for indoor agriculture as they can be sterilised at high temperatures [112]. Growing mediums absorb nutrients and water to help the plants grow. The potential to test for fungal pathogens after watering using a filtration system is yet to be determined.

Chemical treatments to control fungal diseases in crops depends on the plant species being grown and disease in question [113]. Many producers would have to wait until signs of fungal disease damage appeared on crops to be able to identify the causal pathogen species then apply specific fungicide chemistry accordingly. Some growers spray fungicides intermittently or weekly to prevent an outbreak. These sprays are generally synthetic and can be toxic to natural habitats. Fungicide applications can result in long-term toxic residues in water systems, mammals, including humans, and affect other microorganisms such as bacteria and algae [113,114,115]. One study analysed multiple fungicide groups, including but not limited to Benzimidazole, organophosphate and chlorophenyl in aquatic systems and found that substances from these groups had a detection frequency up to 96% in water catchments dominated by agriculture [114]. Another study found that, when feed was contaminated with a fungicide from the ethylene(bis) dithiocarbamate family, the chance of various types of tumour significantly increased [116]. Fungicides are divided into two broad classes, contact and systematic, based on their chemical components and mode of action [115,117]. The contact group focusses on preventing the fungi or fungal spores from developing or growing on plant tissue [115]. The systematic class is absorbed and translocated around the plant to infection sites [115]. Some major classes of fungicide include the methyl benzimidazole carbamates (MBCs), the demethylation inhibitors (DMIs), the quinone outside inhibitors (Qols) and the succinate dehydrogenase inhibitors (SDHIs) [118]. America, Europe and Australia are major agricultural producers and significant users of fungicides, employing these chemicals to protect crops from fungal diseases that can severely impact yields and quality [119]. However, the extensive use of fungicides in these regions has led to the banning of certain chemicals. Banned fungicides are chemicals that have been prohibited due to their harmful effects on human health, wildlife and the environment. In Europe, substances like Chlorothalonil and Mancozeb have been banned because of their potential to cause cancer and environmental damage, particularly to aquatic ecosystems [120,121]. Similar action has been taken in regions like the U.S. and Australia with the prohibited use of the fungicide Carbendazim, with regulatory bodies continuing to monitor and restrict fungicides with harmful side effects [114,122]. These bans reflect an increasing global emphasis on safer agricultural practices and environmental protection. Fungicides can be further broken down to single or multi-site specific due to their modes of action. Single site fungicides such as DMIs and Qols are used to disrupt a single metabolic process or structure and are more prone to the development of resistance in rapidly evolving pathogen populations. Multi-site specific fungicides, as the name suggests, target multiple sites within the fungal pathogen [118]. MBCs target the β-tubulin gene and affect mitosis resulting in toxicity in fungal cells. DMIs target the cytochrome P450-dependent sterol 14α-demethylase (Cyp51) gene and affects the cell membrane of fungi [123]. SDHIs and Qols effect the respiration resulting in decreased energy efficiency in the cytochrome bc1 enzyme complex [123]. Though the chemical treatments are effective in recent years there has been an increase in fungicide resistance [124,125]. Resistance can occur when fungal pathogens have been overexposed to a chemical [126]. This overexposure is predicted to increase due to changing climate conditions and seasonal variation leading to higher disease incidence, in turn increasing fungicide resistance [114]. Climate change and seasonal variations are exacerbating plant disease dynamics by creating favourable conditions such as warmer temperatures, altered humidity, and increased CO_2_ levels that enhance fungal growth and sporulation [114,124]. These changes result in longer infection periods and more severe outbreaks, leading to heightened reliance on fungicides. This overuse accelerates the development of fungicide resistance through mechanisms like genetic mutations in target-site genes or increased efflux pump activity, often resulting in cross-resistance across fungicide classes [124]. As resistance grows, crop yields decline, production costs rise, and environmental risks from overapplication intensify [114,124]. Effective mitigation requires integrated disease management strategies, including cultural practices, biological controls and judicious fungicide use, alongside advancements in predictive modelling [118].

The added challenge to resistance is that a single pathogen can become resistant to multiple fungicides. For example, it was reported that *Botrytis cinerea* is resistant to 15 different fungicide classes with resistant mutations appearing in genes such as *Cyp51*, and cytochrome B (*Cyt B*) [105,118,124]. *Sclerotinia* sp. has shown resistance to the fungicides from the SDHI class with mutations in the *SdhB* and *SdhC* target genes that effect components of the succinate dehydrogenase complex affecting respiration [127,128,129].

## 4. Use of Molecular Diagnostics in Fungal Detection and Identification

Molecular diagnostics utilize techniques to analyse genetic material and include PCR, qPCR, sequencing, LAMP and other isothermal assays, providing precise and rapid identification of pathogens. Advantages of molecular diagnostics include increased sensitivity and high specificity to specific pathogens, important in biosecurity and treatment contexts and overcoming some of the limitations of visible-based approaches whereby different fungi cause similar symptoms and/or are difficult to distinguish at the microscopic level.

Molecular diagnostics are techniques used to identify biological markers in the genome, transcriptome or proteome of viruses, bacteria, plants, humans and fungi [130]. DNA sequencing is commonly used to identify regions of the genome that can be used as a unique identifier for a particular species. Mycologists regularly use the *internal transcribed spacer* (*ITS*) operon or cluster of genes that is found in the region between the small subunit and large subunit of nuclear ribosomal DNA as a target for DNA barcoding [131,132]. This target is commonly used because it is present in all fungus and is highly variable, enabling distinction of different fungal species [133,134]. *Cytochrome C oxidase* (*COX*) genes are found in the mitochondria and commonly used as gene targets. Similar to the *ITS* operon the *COX* gene is commonly used for identification purposes due to it being present in many taxa and at high levels due to multiple copies of mitochondrial DNA [135]. Another gene often used from the mitochondria is *cytochrome B* (*Cyt B*) [136]. The *Cyt B* gene is used for species specific detection and has been used in many phylogenetic studies [137,138,139,140,141,142,143]. *SdhB* and *SdhC* are two of the four genes that aid in the succinate dehydrogenase complex [144,145].

For the last 40 years, the most commonly used molecular diagnostic method for fungal pathogens has been Polymerase Chain Reaction (PCR) [146]. PCR detection methods include observing signs and symptoms of the fungal pathogen, taking a sample of the fungus or infected plant tissue and sending it in the mail to the laboratory for identification. In the laboratory the DNA would be extracted, and tests conducted. The type of fungal pathogen and the quantity and quality of the collected material would be dependent on how quickly results could be obtained. In cases where the amount of collected material would not provide adequate DNA yields, methods to grow the pathogen in vitro may be needed, delaying the time to identification. PCR and DNA visualisation use specialised laboratory equipment such as gel electrophoresis and capillary electrophoresis to assess single nucleotide polymorphorisms (SNPs) and single sequence repeat (SSR) markers. For further analysis and confirmation of the pathogen, DNA sequencing and database searches could be used. PCR has been used to aid in the identification of *Alternaria solani* that is a major threat to the potato and tomato industry by targeting the ITS region [147]. Whilst this study was based in the laboratory, strategies to adapt for in-field would be desirable, however, the use of PCR in-field is limited due to need for specialist laboratory equipment, cost of reagents and requirement for trained personal [148]. A variation of PCR called Real Time Polymerase Chain Reaction (real time-PCR) or quantitative Polymerase Chain Reaction (qPCR) couples amplification and detection methods [36]. This assay can quantify the levels of plant pathogen by measuring the time to amplification in real time, essentially giving a precise quantitative relationship between the amount of starting DNA and the quantity of PCR product. The amount of amplification product is visualised using fluorescence and emission is measured during the reaction [36]. Whilst qPCR reduces time, and is suitable for larger-scale diagnostics, its main constraint is the expensive spectrofluorometric thermal cyclers, the reagents and their maintenance [149]. qPCR has been used in plant pathology for the fungal detection in crop biosecurity applications and for the detection of seed borne fungal pathogens such as *Fusarium graminearum* in wheat, *Magnaporthe grisea* in rice and *Phytophthora infestans* in early stage potatoes [74,149,150,151,152,153].

Loop-Mediated Isothermal Amplification or LAMP is a method for rapid and specific amplification of DNA through the use of 4–6 primers making it faster the PCR. A feature of LAMP is the ability to run assays at isothermal or one temperature making it more amenable to field usage. LAMP has been adapted for portable, in-field diagnostics as it can amplify DNA or RNA in short (15–20 min) time-frames due to the formation of loop like DNA structures [33,154]. LAMP assays are more tolerant to crude or dirty DNA making the sample preparation stage easier and quicker [74]. LAMP is measured by its sensitivity or what the least amount of DNA or RNA is required for detection and specificity or how specific is the assay when compared to closely related species.

LAMP can use fluorescence absorbency measured in real time as well as a range of other visualisation methods, including colorimetric LAMP which uses pH-dependent colour change for rapid visualisation of presence/absence of DNA [155]. These approaches are being used for in-field testing of different economically significant pathogenic fungi. LAMP was successfully used to detect *B. cinerea* inoculated rose petals and pelargonium leaf discs in the laboratory [33]. The assay was highly sensitive and no cross reactivity with other fungal pathogens tested such as *Alternaria brassicola*, *Fusarium avenaceum* and *Sclerotinia sclerotiorum* amongst others was observed, however, amplification did occur with the closely related species *B. pelargonii* [33]. Khan et al. (2018) [147] have been able to develop a working LAMP assay in the laboratory for the detection of *A. solani* on potato leaves with a 10-fold greater sensitivity than traditional PCR.

Over time, other isothermal techniques have been developed (Table 1). Currently, many are foccused on the detection of viruses and bacteria for medical applications. There is a growing potential for isothermal techniques such as LAMP to be further developed and optimised in agricultural settings for the rapid detection of fungal pathogens [97]. All techniques have their advantages and disadvantages. The LAMP assay is more sensitive and specific than traditional PCR however it can be easily contaminated [4,73].

Nucleic acid sequence-based amplification (NASBA) is a transcription based method designed to amplify RNA sequences [97]. This method has a decreased chance of contamination however it cannot amplify double stranded DNA [97]. NASBA has been used to identify *Candida* species in human blood using primers targeting the 18S rRNA sequences, with a further 19 different fungi being tested and showing postive detection [156].

Rolling circle ampification (RCA) is a DNA replication method that uses a Padlock probe to identify a single nucleotide polymorphorisms (SNPs) in the genome. A padlock probe uses a phosphorylated enzyme at the 5′ end and a complement 5′-biotinylated enzyme at the 3′-end. This method can be relatively easily expanded to multiple allele or loci identifications using Ligation-RCA, however, due to the high sensitivity requires precautions to avoid contamination and false positives [99,157]. In 2014 a study reliably identified fungal pathogens at the species and subspecies level including but limited to *Aspergillus*, *Scedosporium* spp., *Cyphellophora* spp., *Fusarium* spp. and *Cryptococcus* spp., many of which effect immunocompromised people [99].

Whole genome amplification (WGA) is considered part of the Multiple Displacement Amplification (MDA) methods [97]. WGA is a non-selective amplification method that enables sequencing of DNA from a single cell. This method is easily contaminated by competing DNA contamination or endogenously generated DNA [97]. To date, MDA has been used to amplify less than 10 ng of DNA template routinely creating 10 kb amplicons of *Penicillium paxilli* and the slow growing endophyte *Epichloe festacae* [102,158].

Strand displacement amplification (SDA) is a recently developed technique of WGA that is used to pre-amplify DNA templates before PCR analysis. This method can detect single spores, however, the technique can only amplify DNA not RNA [106]. This method has been tested on single spores of the arbuscular mycorrhizal fungi *Glomus* and *Gigaspora*. The results showed that SDA was able to amplify products 3.8 to 5.4 µg per reaction for the *Glomus* spp. and 5.8 µg for *Gigaspora. rosea* [106].

Helicase-Dependent Amplification (HDA) is similar to PCR but does not require a heat denaturation step, instead uses a single temperature from start to finish [83]. To achieve the separation of single-stranded DNA it uses helicase, polymerases and other enzymes for amplification [97]. HDA was used to successfully detect the fungal pathogen *M. oryzae* in rice seed and when compared with LAMP assays provided similar sensitivity and specificity, showing no amplification from other fungal pathogens [17].

Recombinase Polymerase Amplification (RPA) reaction works by using enzmyes to open strands of dsDNA and amplifies targets with strand substitution activity at a single temperature, usually between 37–42 °C, within 25 min [110]. RPA uses two primers and a low reaction temperature, making it good for in-field diagnostics with the caveat that it requires a complex composition of enzymes and other additives for its reactions [97]. RPA was used in combination with lateral flow strips to detect *Candida albicans* on the human body with a high specificty and limit of detection of 1 CFU/µL proving to be a rapid and sensitive test [110].

## 5. New Technologies for Rapid, In-Field Diagnostics

Emerging technologies such as lateral flow devices, rapid antigen tests (RATs), biosensors, multiplex assays and point-of-care diagnostics are revolutionizing identification processes by enabling faster, more accurate, and accessible detection of pathogens and other targets. For any new technology to be effectively deployed for the reliable diagnosis or detection of fungal pathogens in planta, it must meet specific criteria: tests should be highly efficient, affordable and user-friendly [159]. Lateral flow devices have been in use for many years, with well-established applications in various fields. Such diagnostic methods have been successfully developed further in human health, notably with the creation of Rapid Antigen Tests (RATs) during the COVID-19 pandemic that began in December 2019 [159]. As COVID-19 spread rapidly worldwide, numerous countries collaborated on the development of diagnostic tools, including RATs, which detect viral proteins from live SARS-CoV-2 virus particles [159]. A lateral flow assay is a paper-based technique used for the detection and quantification of specific antigens or antibodies [160]. It works by binding gold nanoparticles to form an antibody-antigen complex that moves along a test strip, where it binds to additional antibodies at the test line [160]. The accumulation of nanoparticles at the test line produces a visible fluorescent or coloured line [81,160]. The pregnancy test is one of the most common lateral flow assays and has been in use for decades [160]. Despite criticisms regarding RATs’ sensitivity and susceptibility to false positives, they remain simple, user-friendly devices applicable across various sectors, including healthcare, agriculture, food safety and environmental monitoring [160]. In the realm of fungal detection, lateral flow devices are primarily used for medically important fungal pathogens, such as the *Aspergillus galactomannan* and *Scedosporium* species [79,81]. Companies such as Pocket Diagnostic (https://www.pocketdiagnostic.com/, accessed 25 October 2024) and AgDia (https://www.agdia.com/, accessed 25 October 2024) have developed lateral flow devices for the detection of the fungal pathogen *Phytophthora*, however, review articles of the success have been unable to be sourced [161,162].

Another area gaining traction in molecular diagnostics are biosensors [163]. A biosensor is a device which analyses and converts a biological response into a processable signal [164]. A biosensor takes samples such as human, food, cell cultures and environmental samples, incorporates transducers such as bioreceptors (enzymes, antibodies and nucleic acids) and combines them with an electrical interface such as nanoparticles or electrodes, which then converts into an electronic system [163]. Most of the work in the area has been for the benefit of human health in the early detection of many ailments including cancer, neurodegenerative disorders and viral infections [165]. There are four classifications of biosensors depending on the signal system or transducer; these are electrochemical, optical, piezoelectric or thermometric [84]. Electrochemical biosensors combined with the use of fluorescence imaging has proven a viable option for fungal detection as was proven by [166] who were able to detect *Yarrowia lipolytica*, a type of yeast in fuel [167,168]. Electrochemical biosensors have the advantage of a low limit of detection, high specificity and ease of use and ability to produce devices for POC field-based analysis [168]. Imaging for fungal pathogens is a non-invasive plant disease detection method and when combined with the low limit of detection of the electrochemical biosensors has the potential to discover the presence of fungal infections before disease symptoms manifest in the host [169]. Currently there is limited information on the combination on electrochemical biosensors and imaging for fungal detection.

The next generation of disease detection will likely involve real-time detection of multiple fungal species simultaneously, referred to as multiplexing [73]. The technology is based on using one reaction with multiple primer pairs to simultaneously amplify multiple pathogens [74]. Visualisation of multiplex PCR currently requires electrophoresis that generate the amplicons by separating the bands in the same column [74]. The current issue with multiplexing PCR is decreased sensitivity [74]. The most common form of multiplexing is through PCR however advancement with LAMP has proven successful in identifying multiple pathogens [170]. The multiplex LAMP (mLAMP) has been used in the detection of *Pyricularia oryzae* and *Triticum* species. These fungal pathogens cause blast diseases in rice and cereals and showed similar results when the mLAMP was compared to the individual LAMP analysis [170]. To utilise LAMP as a multiplexing technique requires the introduction of an endonuclease recognition site to the LAMP primers that generate amplicons that are specific to the target pathogen [170]. Current methods under investigation for mLAMP are approaches to visualise the result. Currently, there are two options: gel electrophoresis and the addition of post-amplification dyes [170]. Further developments in colorimetric LAMP assays is the hue-saturation-value (HSV) colour model. HSV is a variation of the Red, Green and Blue (RGB) colour model, and describes each colour in terms of its shade (hue and saturation) and brightness (value) [155]. This is an image processing quantitative analysis method that can be graphed using the results from the colorimetric assay and a mobile phone [155].

The point of the care diagnostics area has enormous potential to advanced molecular diagnostics. The agriculture industry would benefit from further development in the areas of field-based multiplex assays for the pathogen detection, and even add the potential for fungicide resistance testing [83]. This, coupled with eDNA sampling and swab-based methods where growers can conduct a very quick and cheap on the spot test, would revolutionise the agriculture industries fight against fungal pathogens.

## Figures and Tables

**Figure 1 jof-10-00851-f001:**
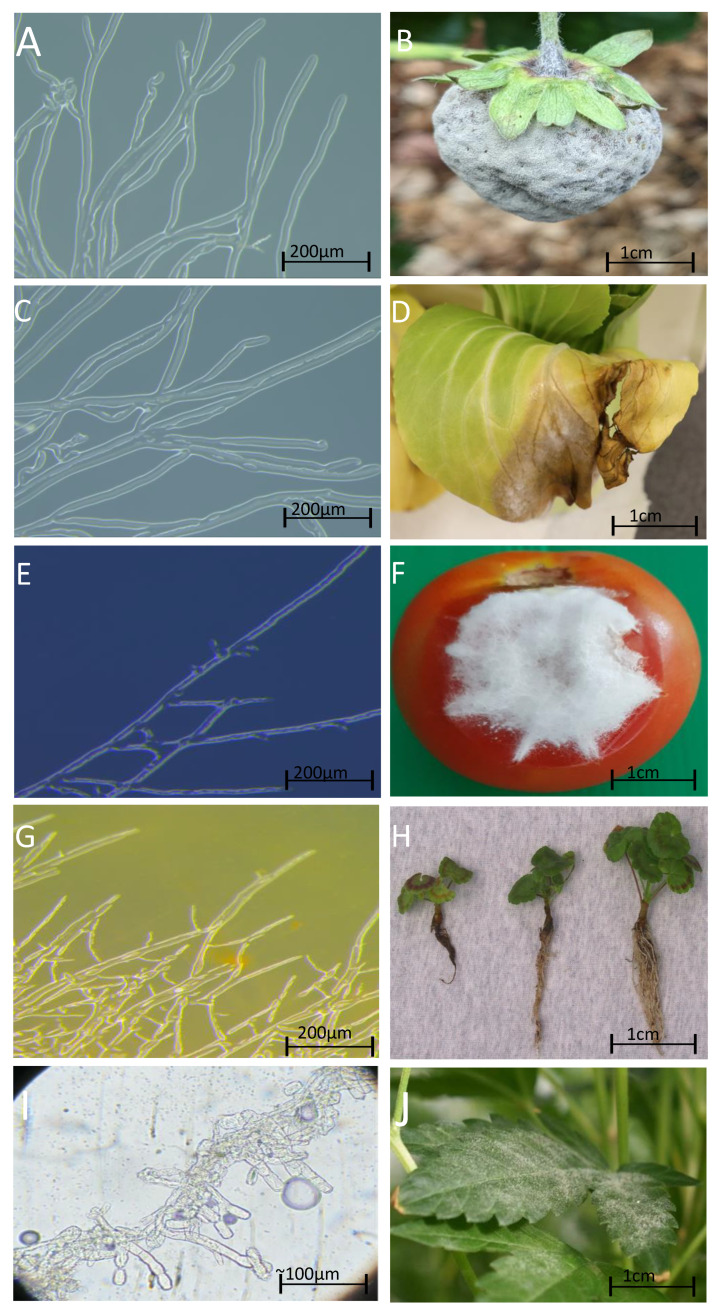
Microscopic and macroscopic images of fungal pathogens detected in protected cropping environments. (**A**) Laboratory-grown *Botrytis cinerea* on a potato dextrose agar plate and viewed under a Leica M205 FA Stereomicroscope. (**B**) Strawberry fruit infected with *Botrytis cinerea* in field (photo taken by Marlo Molinaro La Trobe University). (**C**) *Sclerotinia sclerotiorum* grown in the laboratory on a potato dextrose agar plate and viewed under a Leica M205 FA Stereomicroscope. (**D**) Bok choy leaf infected with *Sclerotinia sclerotiorum*, 8 days after inoculation. (**E**) Laboratory grown putative *Fusarium oxysporum* collected from CEA, grown on a potato dextrose agar plate and viewed under a Leica M205 FA Stereomicroscope. (**F**) *Fusarium* growing on a tomato, reprinted with permission from [30]. Copyright 2021 Zahir Shah Safari from Leibniz Universität Hannover. (**G**) Laboratory grown *Pythium irregulare* now *Globisporangium irregulare* on a potato dextrose agar plate viewed under a Leica M205 FA Stereomicroscope, obtained from Dr. Niloofar Vaghefi from University of Melbourne. (**H**) *Pythium irregularre* now known as *Globisporangium irregulare* growing on geraniums taken and unchanged from CABI Plantwise Plus website (https://plantwiseplusknowledgebank.org/doi/10.1079/PWKB.Species.46152, accessed on 16 October 2024) under a Attribution-NonCommercial-NoDerivatives 4.0 International (CC BY-NC-ND 4.0) license taken by Michael Evans, University of Arkansas. (**I**) Suspected *Golovinomyces cichoracearum* grown on onion viewed under a microscope, taken from the Global Biodiversity Information Facility website (https://www.gbif.org/occurrence/4891810393, accessed on 16 October 2024) by Schmidt Dávid (licensed under http://creativecommons.org/licenses/by-nc/4.0/, accessed on 17 October 2024). *Golovinomyces* is an obligate fungus and therefore unable to be cultured in the laboratory on a growing media. (**J**) Suspected *Golovinomyces cichoracearum* grown on *Cannabis sativa* L. courtesy of industry partner. Scale = 200 µm on (**A**,**C**,**E**,**G**), 1 cm on (**B**,**D**,**F**,**H**,**J**) and ~100 µm for (**I**).

**Table 1 jof-10-00851-t001:** Current list of isothermal based assays used in closed environment agriculture for fungal detection.

Molecular Assay	Summary	Fungal Pathogen/s Detected	Genes Targets	References
Loop-mediated isothermal amplification (LAMP)	Uses 4–6 primers for quick amplification. Performed at 60–68 °C	*Sclerotinia sclerotiorum* *Fusarium oxysporum* *Botrytis cinerea*	Suppressor Of Cytokine Signaling 5 (Ssos5) Intergenic spacer spacer (IGS)	[4,73,95,96]
Nucleic acid sequence-based amplification (NASBA):	Amplifies mRNA by using RNA polymerase, performed at 41 °C	*Candida* sp. *Cryptococcosis*	T7 promoter Capsular-associated protein (CAP10)	[4,23,97,98]
Rolling circle ampification (RCA)	Enzymatic assay that generates ssDNA. Performed between 25 °C and 37 °C	*Candida* sp. *Fusarium* sp.	Translation elongation factor 1-alpha (TEF-1α) Internal transcribed spacer (ITS)	[74,99,100,101]
Whole genome amplification (WGA)	Amplifies entire genome. Has no temperature modulated denaturation	*Penicillum paxilli* *Epichloe festucae* *Pythium insidiosum*	No gene target Requires DNA of 1–10 copies for initial input	[97,102,103,104]
Strand displacement amplification (SDA)	Performs enzymatic site specific nicks at 37–50 °C	*Erysiphales* spp.	β-tubulin1 (Tub1) Internal transcribed spacer (ITS)	[101,105,106,107]
Helicase dependent amplification (HDA)	Uses a thermostable helicase enzyme. Performed at 65 °C	*Pythium insidiosum*	Cytochrome c oxidase subunit 2 (COX2) BNI1 product (Bni1p)	[104,108,109]
Recombinase Polymerase Amplification (RPA)	Invades the double stranded DNA using enzymes and is performed at 30–42 °C	*Bipolaris sorokiniana* *Candida albicans*	Internal transcribed spacer (ITS) Calmodulin (cal)	[107,110,111]

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
