# Peer review of "Isothermal Detection Methods for Fungal Pathogens in Closed Environment Agriculture"

_jof, 2024, doi:10.3390/jof10120851_

Round 1

Reviewer 1 Report

line number 105: please, correct the name and remove dot at the end

line number 138: give more information on that

line number 141: please correct, figure 1J ?

175: please correct, not italic downy mildew

263: I would suggest being more precise regarding the groups of fungicides available worldwide and highlighting differences in legislation. This is important because some fungicides are no longer available in certain countries while remaining in use in others. Please refer to the FRAC website for a comprehensive overview and consider citing it. 

272: Please add something more about this argument, fungicide resistance is a big issue and should be underlined

321: you can not say that qPCR is similar to LAMP, because they are very different, qPCR is a quantitative method , LAMP is a qualitative one. Please be more precise on that!

Dear Author, 

the work is interesting, but I suggest major revision, because, the review needs to be more precise. I suggest it as a mini-review, not a full review. 

Author Response

  1. Comment: line number 105: please, correct the name and remove dot at the end

Response: Thank you for picking up this error. The name has been corrected and the dot at the end removed, now reading ‘H) Pythium irregularre’.

  1. Comment: line number 138: give more information on that

Response: Thank you for the comment. A brief sentence has been included on the differences between the two pathogens and how to view these differences as follows:

‘Closely related powdery mildew species, such as Golovinomyces and Blumeria spp. are difficult to distinguish morphologically, with the main differences being host range and conidial and hyphal structures, which can only be seen under a microscope [42,43].’

  1. Comment: line number 141: please correct, figure 1J ?

Response: Thank you picking up this error. The italics on “1” has been removed.

  1. Comment: 175: please correct, not italic downy mildew

Response: Thank you picking up this error, downy mildew is now unitalicized.

  1. Comment: 263: I would suggest being more precise regarding the groups of fungicides available worldwide and highlighting differences in legislation. This is important because some fungicides are no longer available in certain countries while remaining in use in others. Please refer to the FRAC website for a comprehensive overview and consider citing it.

Response: Thank you for the comment. As per the comment we have added further information about banned fungicides across major agriculture producing countries as follows:

‘America, Europe, and Australia are major agricultural producers and significant users of fungicides, employing these chemicals to protect crops from fungal diseases that can severely impact yields and quality [103]. However, the extensive use of fungicides in these regions has led to the banning of certain chemicals. Banned fungicides are chemicals that have been prohibited due to their harmful effects on human health, wildlife, and the environment. In Europe, substances like Chlorothalonil and Mancozeb have been banned because of their potential to cause cancer, and environmental damage, particularly to aquatic ecosystems [104,105]. Similar action has been taken in regions like the U.S. and Australia with prohibited use of the fungicide Carbendazim, with regulatory bodies continuing to monitor and restrict fungicides with harmful side effects [98,106]. These bans reflect an increasing global emphasis on safer agricultural practices and environmental protection.’

  1. Comment: 272: Please add something more about this argument, fungicide resistance is a big issue and should be underlined

Response: Thank you for the comment. Further information about fungicide resistance and their effects was added to the text to emphasise the effects on crop production:

‘Climate change and seasonal variations are exacerbating plant disease dynamics by creating favourable conditions such as warmer temperatures, altered humidity, and increased COâ‚‚ levels that enhance fungal growth and sporulation [98,109]. These changes result in longer infection periods and more severe outbreaks, leading to heightened reliance on fungicides. This overuse accelerates the development of fungicide resistance through mechanisms like genetic mutations in target-site genes or increased efflux pump activity, often resulting in cross-resistance across fungicide classes [109]. As resistance grows, crop yields decline, production costs rise, and environmental risks from overapplication intensify [98,109]. Effective mitigation requires integrated disease management strategies, including cultural practices, biological controls, and judicious fungicide use, alongside advancements in predictive modelling [102].’

  1. Comment: 321: you can not say that qPCR is similar to LAMP, because they are very different, qPCR is a quantitative method , LAMP is a qualitative one. Please be more precise on that!

Response: Thank you for the comment. Removed “Similar to qPCR”

Detail comments:

the work is interesting, but I suggest major revision, because, the review needs to be more precise. I suggest it as a mini review, not a full review.

Response: We believe by incorporating both reviewers feedback we have made the review more detailed and precise.

Reviewer 2 Report

This article reviewed recent advancements in technology, particularly in molecular and isothermal diagnostics, offer promising tools for the early detection and management of fungal pathogens and explored isothermal amplification and other new technologies in detection of fungal pathogens that occur in CEA. However, there some revisions should be made in the manuscript, as following:

1. The format of the paper needs to be modified according the requirement of the journal. Like table 2.

2. in the part of “Molecular diagnostics”, the authors should summarize and comment relevant technologies instead of listing them simply.

3. The authors should clarify the difference of the technologies among the parts of “Fungal identification and detection methods”, “Molecular diagnostics” and “New technologies to aid identification”.

No

Author Response

Major comments:

This article reviewed recent advancements in technology, particularly in molecular and isothermal diagnostics, offer promising tools for the early detection and management of fungal pathogens and explored isothermal amplification and other new technologies in detection of fungal pathogens that occur in CEA. However, there some revisions should be made in the manuscript, as following:

  1. Comment: The format of the paper needs to be modified according the requirement of the journal. Like table 2.

Response: Thank you for the comment. As suggested, the format of the table has been changed to meet the journal requirements.

  1. Comment: in the part of “Molecular diagnostics”, the authors should summarize and comment relevant technologies instead of listing them simply.

Response: Thank you for the comment. A list of molecular approaches is mentioned in the Fungal identification and detection methods section in the context of the breadth of approaches used to identify fungus. These approaches are outlined in more detail in the following ‘Molecular diagnostics’ section where the main focus was on qPCR and LAMP, as these are widely used and highly relevant techniques. We also included a brief description of other isothermal methods and included a table outlining key components of each assay. We believe we have provided detailed explanations of the key techniques while highlighting the importance of other methods to offer a comprehensive perspective.

  1. Comment: The authors should clarify the difference of the technologies among the parts of “Fungal identification and detection methods”, “Molecular diagnostics” and “New technologies to aid identification”.

Response: Thank you for the comment. We have changed the title of the sections and added an additional introductory sentence outlining the focus of each section and aid in clarifying the differences in how the technologies have been used. The “Fungal identification and detection methods” has been changed to ‘Fungal identification and treatment in agricultural settings’ to more accurately represent that this section provides an overview and historical perspective of approaches to identify fungal pathogens and the approaches growers use to monitor and treat outbreaks. The ‘Molecular diagnostics’ section has been renamed ‘Use of molecular diagnostics in fungal detection and identification’ and introductory paragraph added as follows:

‘Molecular diagnostics utilizes techniques to analyse genetic material and includes PCR, qPCR, sequencing, LAMP and other isothermal assays, providing precise and rapid identification of pathogens. Advantages of molecular diagnostics include increased sensitivity and high specificity to specific pathogens, important in biosecurity and treatment contexts and overcoming some of the limitations of visible-based approaches whereby different fungi cause similar symptoms and/or are difficult to distinguish at the microscopic level.’

The ’New technologies to aid identification has been renamed ‘New technologies for rapid, in-field diagnostics’ to emphasise that this section will focus on adaptation of point-of-care used largely in medicine to agricultural settings. 

Round 2

Reviewer 1 Report

I do not have any comments, this version is ok and could be submitted. 

I do not have any comments, this version is ok and could be submitted.